# Mechanical Properties of Brass under Impact and Perforation Tests for a Wide Range of Temperatures: Experimental and Numerical Approach

**DOI:** 10.3390/ma13245821

**Published:** 2020-12-21

**Authors:** Maciej Klosak, Tomasz Jankowiak, Alexis Rusinek, Amine Bendarma, Piotr W. Sielicki, Tomasz Lodygowski

**Affiliations:** 1Laboratory for Sustainable Innovation and Applied Research, Technical University of Agadir, Technopole d’Agadir, Qr Tilila, Agadir 80000, Morocco; b.amine@e-polytechnique.ma; 2Institute of Structural Analysis, Poznan University of Technology, Piotrowo 5 St., 60-965 Poznan, Poland; tomasz.jankowiak@put.poznan.pl (T.J.); piotr.sielicki@put.poznan.pl (P.W.S.); 3Laboratory of Microstructure Studies and Mechanics of Materials LEM3, University of Lorraine, UMR-CNRS 7239, 7 rue Félix Savart, 57073 Metz, France; alexis.rusinek@univ-lorraine.fr; 4Institute of Combustion Engines and Powertrains, Poznan University of Technology, Piotrowo 3 St., 60-965 Poznan, Poland; tomasz.lodygowski@put.poznan.pl

**Keywords:** gas gun experimental technique, high rates of loading, brass properties, numerical simulations, failure criterion

## Abstract

The originally performed perforation experiments were extended by compression and tensile dynamic tests reported in this work in order to fully characterize the material tested. Then a numerical model was presented to carry out numerical simulations. The tested material was the common brass alloy. The aim of this numerical study was to observe the behavior of the sample material and to define failure modes under dynamic conditions of impact loading in comparison with the experimental findings. The specimens were rectangular plates perforated within a large range of initial impact velocities V_0_ from 40 to 120 m/s and in different initial temperatures T_0_. The temperature range for experiments was T_0_ = 293 K to 533 K, whereas the numerical analysis covered a wider range of temperatures reaching 923 K. The thermoelasto-viscoplastic behavior of brass alloy was described using the Johnson–Cook constitutive relation. The ductile damage initiation criterion was used with plastic equivalent strain. Both experimental and numerical studies allowed to conclude that the ballistic properties of the structure and the ballistic strength of the sheet plates change with the initial temperature. The results in terms of the ballistic curve V_R_ (residual velocity) versus V_0_ (initial velocity) showed the temperature effect on the residual kinetic energy and thus on the energy absorbed by the plate. Concerning the failure pattern, the number of petals N was varied depending on the initial impact velocity V_0_ and initial temperature T_0_. Preliminary results with regard to temperature increase were recorded. They were obtained using an infrared high-speed camera and were subsequently compared with numerical results.

## 1. Introduction

Dynamic tests are rarely coupled with thermal analysis since it is not easy to reach uniform temperature in the plate and gas gun is not frequently provided with a thermal chamber. This complex technique is relatively new. The usual approach is to carry out perforation tests at room temperature and to extrapolate results using numerical simulations at high temperatures knowing the constitutive relation. Many authors dealt with perforation analysis, from theoretical approaches such as those discussed in [1,2], aimed to define material ballistic properties through relations between impact and residual velocities with the ballistic limit, to more practical considerations as reported in [3,4,5,6,7,8] where numerical applications are discussed for a variety of materials like metals or concrete. However, no direct data concerning perforation failure modes at high temperatures were reported or published. The thermal softening of the material was usually tested using quasi-static experiments and its extrapolation to high strain rates was often a rough simplification. 

Thus, running dynamic experiments with the use of a thermal chamber is a solution to avoid these limitations. Such experiments were carried out using brass and are widely discussed in [9]. Brass alloy was selected for this work since this material presents no or small strain rate sensitivity [10]. More results using this new heating system were already provided for other materials including metals or polymers [11,12,13,14].

The new own dynamic tests results presented here enabled us to characterize the material and to define the constitutive relation. Standard compression and tension tests resulted in stress–strain curves and provided constants to the selected constitutive relation. They also determined failure strains which were taken into consideration in the numerical model. The tests covered moderate strain rates of up to 2200 1/s and the range of temperatures from the room temperature to 453 K. Some experimental results using infrared high-speed camera are also reported in order to study the thermal response of the material and to compare it with numerical estimations.

The Abaqus/Explicit was used to simulate all required impact velocities V_0_ and initial temperatures T_0_. The parametrical analysis of the failure criterion was carried out to demonstrate its temperature and strain rate dependence. It consisted of observing the petaling failure mode and fitting the numerical ballistic curves to the experimental ones in order to propose the failure criterion value. 

## 2. Experimental Approaches 

The new experimental results of unidirectional tests of tension and compression, allowing to estimate the material constants and to ensure accurate FE simulations, are presented in this chapter. They are complimentary to a number of dynamic perforation tests previously conducted at high impact velocity [9] in order to understand the thermo-mechanical behaviour of brass and to evaluate its mechanical properties. 

### 2.1. Tension under Quasi-Static Conditions

The quasi-static uniaxial tensile tests were performed using a conventional hydraulic machine ZWICK (Metz, France) dedicated for tensile/compression tests. The specimen had a typical dumbbell-shaped specimen of 16 cm in length and 1 mm in thickness. One end of the specimen was embedded on 40 mm while the other one was fixed to the mobile crosshead. The loading force and the displacement were recorded during the tests for each imposed velocity.

The available strain rates were 0.001 to 0.01 1/s and the initial temperature T_0_ was varying between 293 K and 453 K. The results relating to stress–strain curves are presented in Figure 1a.

A very slight negative thermal effect is observed in shifting the curve downwards, whereas no strain rate effect can be spotted. The temperature effect is visible as far as the failure strain is concerned. For the strain rate of 0.001 1/s, the true strain at failure decreases from 0.37 to 0.32 when the temperature increases from 293 K to 453 K. For the strain rate of 0.01 this effect is greater in nominal values, and changes from 0.445 to 0.375.

### 2.2. Compression under Dynamic Loading Using Sandwich Configuration

Tensile tests were carried out using standard cylindrical specimens of 6 mm in diameter and 4 mm in thickness (4 glued pastilles of 1 mm each—see Figure 2). The glued four-layer sandwich specimen was analyzed in [15,16] in order to see if the interface considerably changed the macroscopic results compared to a bulk material; the friction value effect is estimated to be approximately μ = 0.2. As the material used in this work does not display a strength differential effect, the technique used may be applied and thus, the tests may be performed using the Split Hopkinson Pressure Bars (Metz, France) [17,18].

Compression tests were carried out at room temperature for four selected impact velocities corresponding to three different strain rates. Two tests were proposed for each initial impact velocity V_0_, and the results are recapitulated in Figure 1b. The first conclusion is that brass demonstrates a relatively small impact of the strain rate effect on flow stress level. It is visible that failure occurred faster for lower impact velocities. It seems this is due to brass ductility which is most pronounced at higher strain rates where the thermal effect is stronger. The same was observed in tensile tests: for room temperature the strain at failure equaled to 0.375 at 0.001 1/s and became 0.44 when the strain rate was increased 10 times.

### 2.3. Perforation Tests

As the perforation tests are detailed enough in [9], they will not be discussed in detail in this paper. Only a summary of all data is given in Table 1. The gas gun set-up and specimen dimensions are presented in Figure 3, Figure 4 and Figure 5. Those perforation tests were performed for a wide range of temperatures, from 293 K to 533 K. In addition, a wide range of initial impact velocities V_0_ from 40 to 120 m/s was used during the tests. 

The gas gun set-up is shown in Figure 3. The specimen and the projectile dimensions are presented in Figure 4. A conical projectile with an angle of 72° and 11.5 mm in diameter was used in the tests. The target plate is 0.8 mm thick with a size of 130 × 130 mm^2^ which is clamped along its perimeter to provide a perfect fixation and to avoid sliding. 

The apparatus is equipped with a thermal chamber “E” in which a specimen “I” is placed. The temperature is changed from room temperature to a maximum temperature of 533 K.

The air flows inside the system through to a ventilator. A sarcophagus casing is used around the plate specimen “I” to maintain uniform temperature distribution. Therefore, the two sides of the specimen are heated up at the same time. Due to conductivity, the entire specimen reaches the initial temperature imposed to the specimen and is regulated by the PID controller “H”. 

A schematic overview of the oven used during perforation and the heat airflow is shown in Figure 5. The electronic heat controller (PID type) permits to set the required temperature and to control temperature increase. The temperature values in the thermal chamber and in the specimen are measured by two thermocouples. The built-in thermocouple measures continuously the temperature inside the chamber. Before the analysis, a special calibration specimen with an integrated thermocouple is used to measure the temperature evolution in the specimen itself which allows to set up the parameters of the PID controller. These measurements are important as the temperature imposed by the user and regulated by the controller does not exactly match the temperature in the specimen. A difference is observed between these two measurements due to the heat loss due to conductivity of the device. It corresponds to 28–30% of the temperature assumed. Therefore, a waiting time t_waiting_ ≈ 20 min is necessary to obtain the required uniform temperature distribution along the specimen. The procedure of calibration must be repeated if the material is changed or if the thickness of the material studied is modified. 

The projectile is launched using a pneumatic gas gun which accelerates in the tube to reach the initial impact velocity V_0_. Then, the projectile impacts the brass plate with partial or complete perforation depending on the quantity of kinetic energy transferred to the tested plate. The residual velocity V_R_ of the projectile is measured after perforation. A pair of laser sensors are used to measure the initial impact velocity and a laser barrier is fixed behind the plate to measure the residual velocities. The projectile mass is equal to 30 g. The material used for machining the projectile is a maraging steel with a heat treatment to reach a yield stress close to 2 GPa in order to avoid a mushroom effect. Therefore, the projectile has no visible permanent deformation during the process of perforation and may be defined as a rigid body during numerical simulation. 

The residual velocity of the projectile V_R_ may be fitted using the following equation proposed by Ipson and Recht [3], using the shape parameter based on experiments and knowing the ballistic limit velocity V_B_:(1)$$V_{R}={\left({V_{B}}^{\kappa }-{V_{B}}^{\kappa }\right)}^{\frac{1}{\kappa }},$$
where V_0_ is the initial impact velocity. In the above equation V_B_ is equal to 40 m/s and κ = f (T) is varying from 1.85 at T_0_ = 293 K to 2.29 at T_0_ = 533 K. The parameters of Equation (1) are calculated using the least squares method based on experimental results. The energy absorbed by the plate E_d_ can be calculated using the following equation:(2)$${E_{d}=\frac{m_{p}}{2}\left(V_{0}{}^{2}-V_{R}{}^{2}\right)|}_{T}$$
where m_p_ is the projectile mass, the residual impact velocity V_R_ = f(T) and temperature T are fixed for a given calculation and depend on the initial temperature used to perform the impact and perforation tests.

The initial and residual impact velocities are given in Table 1.

The extended experimental analysis provided necessary data to describe the analysed material using an efficient constitutive relation. The next paragraph section the equation used in numerical simulations.

## 3. Modelling of the Thermoviscoplastic Behaviour

Brass is a substitutional alloy composed mainly of two main pure elements, i.e., copper and zinc. Changing of the proportions of copper and zinc alters the properties of the alloy. In this study, a common industrial alloy has been used to carry out the tests. Different phenomenological constitutive relations have been considered to model the material behaviour [19,20,21,22,23,24,25].

Following the analysis of the presented compression/tension tests, the simple phenomenological equation proposed by the Johnson–Cook model [19] was applied in order to reflect the material behaviour. This thermo-viscoplastic behaviour of brass alloy is described by Equation (3) and was applied in the numerical simulations.
(3)$$\sigma =\left(A+{B\varepsilon }_{\mathrm{pl}}^{n}\right)\left(1+\mathrm{Cln}\frac{{\dot{\varepsilon }}_{\mathrm{pl}}}{{\dot{\varepsilon }}_{0}}\right)\left(1-T^{*m}\right)$$
where A is the yield stress, B and n are the strain hardening coefficients, C is the strain rate sensitivity coefficient, ${\dot{\varepsilon }}_{0}$ is strain rate reference value, and m is the temperature sensitivity parameter. The non-dimensional temperature T* for temperature ranging between T_0_ and T_m_ is defined in the following form:(4)$$T^{*}=\frac{T-T_{0}}{T_{m}-T_{0}}$$
where T_0_ is the reference room temperature and T_m_ is the melting temperature. The parameters adopted are presented in Table 2. They were obtained through the implementation of the Johnson–Cook model to our own experimental curves (dynamic tension using SHPB for different temperatures) using the least squares method.

In order to consider the thermal softening of the material used in this work, the constitutive relation is coupled to the heat equation considering adiabatic conditions.

In order to define the failure mode, the classical Johnson–Cook failure model [20,23] was first studied. As demonstrated by closer analysis, the strain rate and temperature effect on the failure criterion is marginal within the studied range of strain rates and temperatures; therefore, a simplified version was adopted in the form of the constant equivalent plastic strain at failure $\varepsilon_{f}^{\mathrm{pl}}$. The parametrical analysis of the failure criterion consisted of observing the failure modes (petaling) and fitting of the numerical ballistic curves to the experimental ones. As a result, the value adopted for the equivalent plastic strain at failure $\varepsilon_{f}^{\mathrm{pl}}$ is equal to 0.7.

## 4. Numerical Simulations of Perforated Tests

The numerical simulations were carried out using the explicit finite element solver of Abaqus/Explicit. The mesh size sensitivity was considered taking into account the failure pattern and the value of the residual velocity. Thanks to this analysis the optimal mesh is used in all simulations of the perforation problem. The following number of elements was used in the final analysis (3D finite solid elements): For discretization of the specimen: fine mesh in the middle: 226,805 nodes, 180,096 elements C3D8R (5 elements along the thickness, 0.2 mm × 0.2 mm × 0.2 mm); remaining part: 10,540 nodes, 7888 elements C3D8I (2 elements along the thickness, 2 mm × 2 mm × 0.2 mm; both parts of the specimen were tied in the analysis, the refined mesh part has a form of a circle in the region of contact between the two acting bodies;For discretization of the projectile: 9302 nodes including cylinder 960 C3D8R elements and cone 5516 C3D10M (tetra) elements; both parts of the projectile were tied in the analysis.

Incompatible mode elements such as C3D8I are first-order elements that are enhanced by incompatible modes to enhance their bending behaviour. In addition to the standard displacement degrees of freedom, incompatible deformation modes are added internally to the elements. The incompatible mode elements use full integration and, thus, have no hourglass modes. In the presented model, this type of elements was used outside the perforation area where bending occurred and the behaviour was mostly elastic.

The mesh is presented in Figure 6. The projectile itself is assumed as rigid body which reduces calculation time since no mushroom effect was observed during experiments. The specimen is modelled using the Johnson-Cook constitutive relation which is a particular type of Mises plasticity model with analytical forms of the hardening law and rate and temperature dependence. The friction parameter between the projectile and the plate was assumed as constant and equal to 0.2 [6,15]. The general contact was used together considering interior contact surfaces created during the failure or erosion of the mesh related to the material. The applied failure criterion based on the equivalent plastic strain $\varepsilon_{f}^{\mathrm{pl}}$ (PEEQ in Abaqus) eliminates any single element from the global mesh once a critical value of strain imposed is reached. It allows for the easier observation of a typical failure mode in the form of petaling related to perforation using conical projectiles. The typical failure mechanism in case of metal sheet perforation appears if the state of stress in the damaged zone is close to uniaxial tension (triaxiality equal to 1/3). This is the reason why the damage initiation parameter was calibrated based on the uniaxial traction tests and optimized by numerical simulation. The latter was compared with the experimental results for higher temperatures and brought to the conclusion that the optimal value was temperature independent. The damage evolution parameter (plastic displacement at failure) was assumed 0.001 mm, the element size in the impacted zone being equal to 0.2 mm. 

The specimen is fixed on the perimeter assuming a complete embedding. The analysis was assumed as pure mechanical including adiabatic heat effects defined by the Quiney–Taylor coefficient assumed as a constant value of 0.9. The material parameters used are: specific heat C_p_ = 380 J/kgK and density ρ = 8587 kg/m^3^. The initial temperature T_0_ was varying during the calculations. The range of initial temperatures T_0_ covered experimental tests (293 K or 533 K) and was extended to 923 K.

The failure criterion applied in simulations eliminates the elements in which the critical value of the equivalent plastic strain $\varepsilon_{f}^{\mathrm{pl}}$ is reached. It allows for the better observation of a typical failure mode in the form of petaling related to perforation using conical projectiles. The next section will compare numerical findings with experimental results of the perforation tests [9].

## 5. Discussion of Results

The failure modes observed during impact and perforation depend on both impact velocity and initial temperature value. For a conical projectile, a failure mode by petaling occurs inducing radial necking due to the process of piercing [1,6,26,27,28]. The conical projectile perforates the target plate and the plastic strain is localized at the extremities of the petals. Analytical predictions discussed in [23] are fully confirmed at room temperature, whereas more discrepancy in petals number is reported at higher temperatures. 

The usual failure mode is by petaling with three or four petals. Up to six petals have been observed at lower impact velocities (close to the ballistic limit) and at higher temperatures. The failure modes observed during experiments are presented in Figure 7. 

According to the theoretical considerations of Landkof and Goldsmith in [29], the number of petals N strongly depends on the failure strain parameter and mechanical properties such as yield strain and hardening coefficient n. This parameter changes with temperature, i.e., the elongation of the material is more substantial at higher temperatures which directly leads to the increase in the number of petals. This is confirmed in this study: 5 and 6 petals were observed for temperatures over 500 K, whereas 3 or 4 petals were mainly observed at lower temperatures (Figure 7). This behaviour may be explained by a direct relation between the hardening coefficient n and the failure equivalent plastic strain $\varepsilon_{f}^{\mathrm{pl}}=f\left(T\right)$. The smaller the n coefficient, the smaller failure strain value and thus the lower number of petals N. 

The nose angle of the conical projectile also has a direct influence on the failure mode, because it is related to the local stress state from tension to shear and compression in the vicinity of the nose projectile [30,31,32]. However, this parameter is kept constant in this study. 

The perforation process causes an instantaneous increase in the temperature localized in the perforated zone. Plastic deformation energy dissipated during the perforation process is transformed into thermal energy and induces a considerable increase in the local temperature along the petals. The analysis was carried out at room temperature. High-speed infrared camera measurements of the temperature change on the obstacle surface are performed using FLIR SC7300 infrared camera (Poznan, Poland). The calibration parameters for the standoff distance of 500 mm from the obstacle allowed the application of a constant emissivity parameter equal to 0.22, neglecting extender rings and equipment recalibration. Typically, the use of extender rings between the thermal camera and the optics forces is a compromise between the resolution quality, image degradation and temperature accuracy. In the present study, the rings were not used. The constant emissivity value was calibrated specially for the brass specimens based on simple tests, i.e., comparison of high speed camera measurements and credible laser measurement for the static loading process using laser noncontact pyrometer. Moreover, a thin layer of white Talc was used for matting, i.e., to reduce reflectivity on the external surface of the brass specimen according to our own procedures and commonly used calibration techniques [33,34,35,36]. 

The camera adjustment and Altair measuring software made it possible to obtain the resulting resolution of images equal to 892 by 838 pixel with more than 3 kHz recording frequency. It was enough to cover all frames in all of the 0.27 ms steps of the perforation process. Finally, the temperature change in time function was found for all external surfaces of brass plate specimens.

The temperature increase ΔT^exp^ observed in the experiment was between 60 K and 90 K which confirms the transient, adiabatic nature of the process. On the other hand, the projectile’s tip reached the temperature increase ΔT of 150 K during perforation. 

Some findings obtained using thermal imaging camera are displayed in Figure 8a, followed by numerical findings presented in Figure 8b.

This important observation is made in the simulations referring to the failure form. Figure 9 shows a typical petaling encountered during penetration. The images are given for one optimal failure criterion and for two initial temperatures T_0_ = 533 K. It can be stated that the most realistic forms of petaling are those obtained at failure strain of 0.7, this was concluded from a more complete parametrical study carried out concerning the failure strain in which the analysed range was 0.3–0.7. However, the smaller values of the analysed failure strain (close to 0.3) resulted in rugged tips of petals which was not the failure mode observed during the tests. In almost all simulation cases, the number of petals N varied between four and five, and the most common number was four, which is fully in accordance with the experimental findings. The material behaved in a ductile way during perforation and a model reflecting this behaviour was developed.

Figure 9 also summarizes the observations concerning temperature increase during perforation. The highest values of temperature increase ΔT^num^ are observed for the failure strain of 0.7, this is max ΔT^num^ = 118 K and max ΔT^num^ = 98 K for initial temperature of T_0_ = 293 K and T_0_ = 533 K, respectively. These values should be considered for direct comparison with experimental findings. The slightly lower ΔT values were numerically recorded for T_0_ = 723 K and T_0_ = 923 K, ΔT^num^ = 74–75 K and ΔT^num^ = 45–46 K, respectively. The conclusion is that the values linked with the critical failure strain of 0.7 are close to reality reported during the infrared high speed camera analysis where ΔT^exp^ was from 60 K to 90 K.

Figure 10 compares analytical, experimental and numerical results at room temperature and for temperature of 533 K. It can be noticed that an increase in the initial temperature of the specimen shifts the ballistic limit (state of no perforation) to lower values: the ballistic limit obtained is approximately 43 m/s (293 K) and 40 m/s (533 K) in case of the conical projectile. Experimental and numerical results coincide with the Recht–Ipson curves. First, the failure criterion of 0.3 was studied but provided erroneous results, the material was too rigid and thus the simulation points were excessively shifted outside the correct zone (to the right). Two failure equivalent plastic strains of 0.6 and 0.7 were then compared to the experimental and the analytical approach. It can be concluded that the equivalent failure strain of 0.7 is adequate to represent the real material behaviour for both temperatures studied.

Figure 11 extends the discussion of the ballistic limit to the temperatures not covered by the experiment, the following initial temperatures T_0_ have been covered by numerical simulations: 293 K, 533 K, 723 K and 923 K. This numerical extrapolation confirms a significant shifting of the ballistic curve, which demonstrates the ballistic limit is reduced by approximately one third between room temperature and 923 K, from 43 m/s to 28 m/s.

Figure 12 compares maximum local strain rates obtained at different initial temperatures T_0_ (293 K, 533 K, 723 K, 923 K) and two initial impact velocities V_0_ (45 m/s, 100 m/s) as indicated by numerical simulations. The maximum local strain rate was estimated as the mean value of data gathered across all elements located at the petals edges. The order of magnitude of the maximum strain rate level remains the same throughout the whole range studied. It can be observed that the maximum strain rate decreases with temperature increase. The local maximum is obtained at V_0_ = 100 m/s and for T_0_ = 293 K for which the value of ~5.9 × 10^3^ 1/s was recorded during simulations and diminished to ~1.9 × 10^3^ 1/s for T_0_ = 923 K when approaching the melting point. The same tendency is observed for the lower initial impact velocities. This might be explained by the softening of material which accelerates perforation and thus does not allow to develop higher strain rate maxima.

As to energy values dissipated during the perforation process, they diminish with impact velocity increase as well as in higher temperatures (6 times between 293 K and 923 K). The values of energy demonstrated in Figure 12 were obtained from experimental measurements of initial and residual velocities (Equation (2)).

An extensive parametrical study of the failure criterion at different plastic deformation triggering failure has brought a conclusion that this critical value is rather constant and its temperature or strain rate dependence cannot be explicitly described for the temperature range studies. It is to be noted that a change in the failure criterion impacts strongly the ballistic response in terms of the ballistic limit as well as the failure mode. The proposed failure value of $\varepsilon_{f}^{\mathrm{pl}}=0.7$ demonstrated a very good agreement to experimental findings.

## 6. Concluding Remarks

The principal aim of the study was to propose an effective model for the numerical simulation of brass specimen. Several experimental techniques, including HSPB, tensile machine and gas gun were first used to study the material in compression, tension and perforation. The tests were performed within a wide range of temperatures, starting from ambient and reaching 533 K. Some of the tests had been conducted previously and reported by the authors in [9], some others are presented herewith.

The recorded experimental data enabled us to discuss the failure mode in perforation, which was the main point of interest in FE analysis. The other results were used to determine stress–strain curves and consequently constants for the constitutive equation proposed for brass. The Johnson–Cook phenomenological model turned out to be an efficient solution to simulate brass behaviour. The failure criterion adopted allowed for the precise comparison between experiment and simulation. The failure equivalent plastic strain parameter was $\varepsilon_{f}^{\mathrm{pl}}=\;0.7$. Its optimal value is temperature and strain rate independent. The maxima local strain rates recorded in simulations were of the order of 6 × 10^3^ 1/s depending on the initial impact velocities.

The applied FEM model using 3D solid elements gave satisfactory results. In order to limit the volume of the numerical task, only the perforated zone had a refined mesh. Yet, the numerical model contained 194,460 elements and 246,647 nodes. Solid elements provided better results than preliminary simulations with shell elements described in [9]. Abaqus/Explicit solver was used in all simulations.

The ballistic limit is a function of the projectile shape and temperature. For the conical projectile the value measured is equal to 43 m/s at room temperature and it diminished to 40 m/s at 533 K. This was confirmed by the numerical results. The optimization of the failure criterion facilitated a reliable numerical solution for a wide spectrum of temperatures and strain rates.

Thermal images for the infrared high speed camera permitted to observe local heating during perforation process. The range of temperature increase recorded was ΔT^exp^ = 60–90 K, which corresponds to the numerical findings where the output showed max ΔT^num^ = 98–118 K depending on initial impact velocity V_0_ and initial temperature T_0_.

## Figures and Tables

**Figure 1 materials-13-05821-f001:**
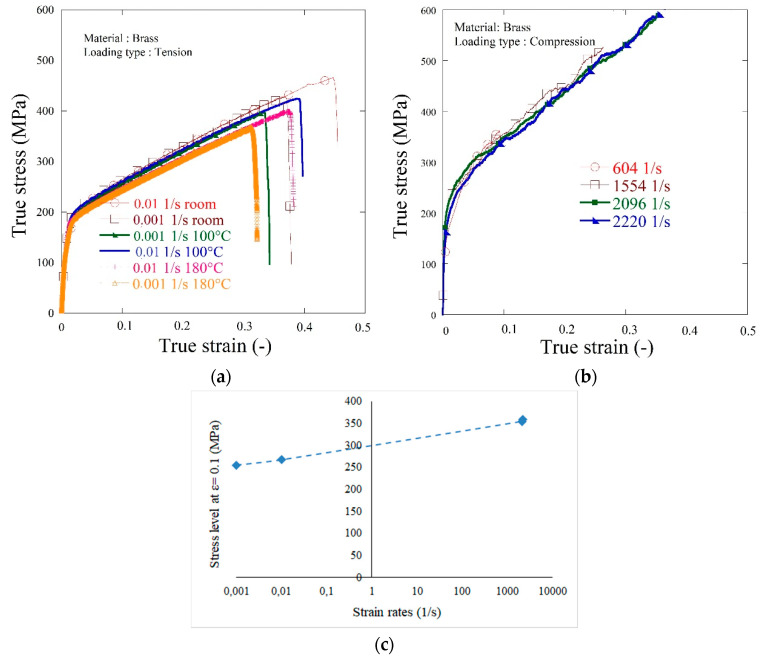
Dynamic experimental results for selected strain rates and temperatures: (**a**) tension tests: true stress vs. true strain curves; (**b**) compression tests: true stress vs. true strain curves; (**c**) strain rate effect on plastic flow.

**Figure 2 materials-13-05821-f002:**
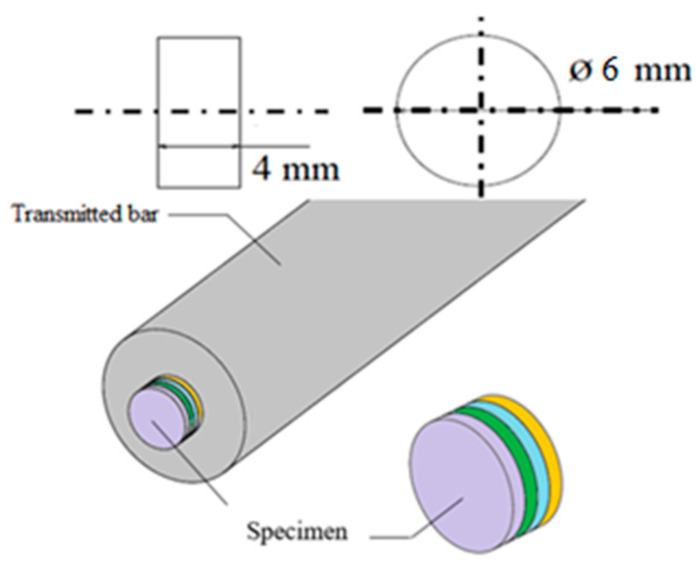
Geometry of test specimen and configuration under dynamic compression [17].

**Figure 3 materials-13-05821-f003:**
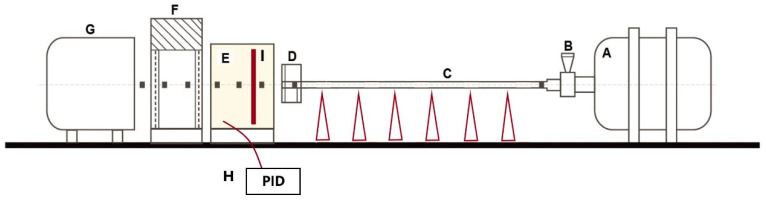
Scheme of gun set-up used for perforation tests at high impact velocities and temperatures. A: pneumatic chamber, B: fast valve, C: gas gun tube with supports, D: sensor for initial impact velocity measurements, E: thermal chamber and specimen fixation device, F: sensor for residual velocity measurement, G: projectile catcher, H: PID controller, I: specimen, ∎∎∎ projectile trajectory.

**Figure 4 materials-13-05821-f004:**
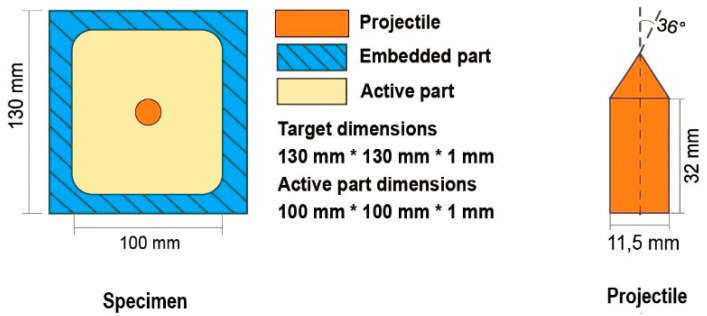
Dimensions of the plate used for perforation and projectile shape.

**Figure 5 materials-13-05821-f005:**
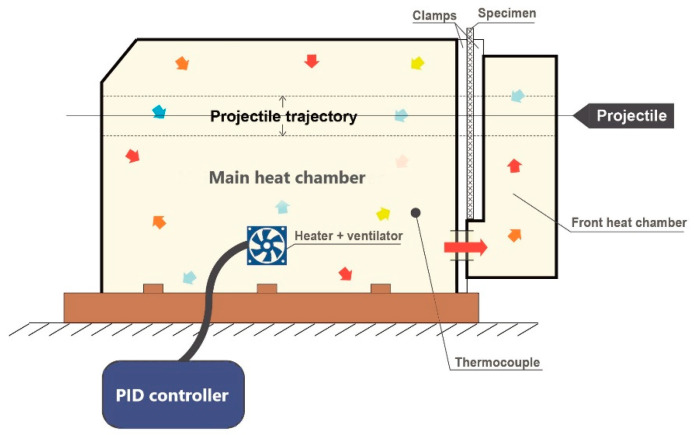
Thermal chamber for heating up the target plate specimens—schematic representation of the process of air mixing.

**Figure 6 materials-13-05821-f006:**
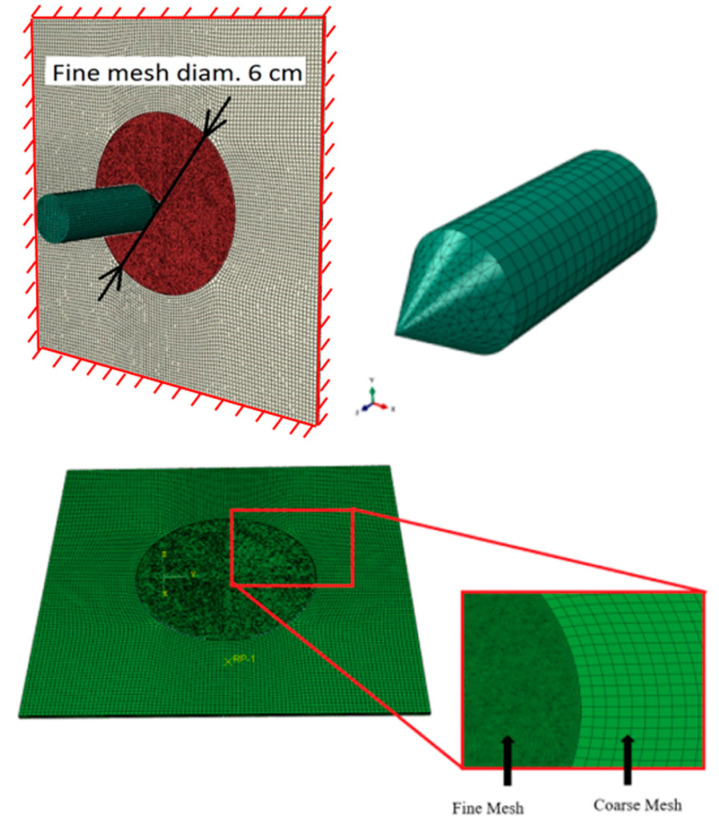
Meshing of projectile and specimen; rough mesh zone (grey colour) and fine mesh zone (burgundy colour, circle diameter 6 cm) applied in the vicinity of impact point; plate thickness 0.8 mm.

**Figure 7 materials-13-05821-f007:**
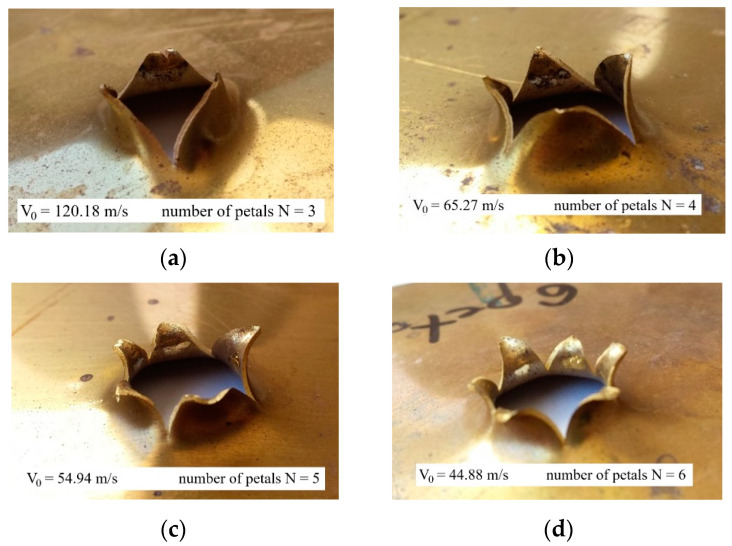
Initial temperature effect and impact velocity during perforation. Experimental observations of petaling failure mode, (**a**) 3 petals at T_0_ = 293 K and V_0_ = 120.18 m/s, (**b**) 4 petals at T_0_ = 373 K and V_0_ = 65.27 m/s, (**c**) 5 petals at T_0_ = 473 K and V_0_ = 54.94 m/s, (**d**) 6 petals at T_0_ = 533 K and V_0_ = 44.88 m/s.

**Figure 8 materials-13-05821-f008:**
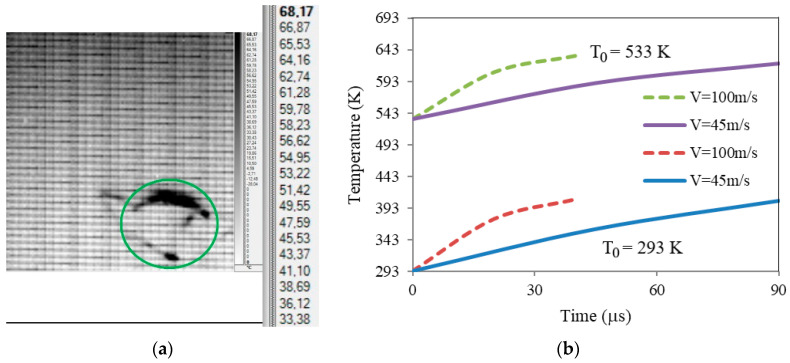
Temperature evolution during perforation process, (**a**) thermal heating captured by thermal imaging camera, initial temperature T_0_ = 293 K (20 °C), impact velocity V_0_ = 89.1 m/s; darker parts indicate local temperature increase, legend displayed in °C [9], (**b**) temperature evolution in most heated elements of petals recorded during numerical simulations.

**Figure 9 materials-13-05821-f009:**
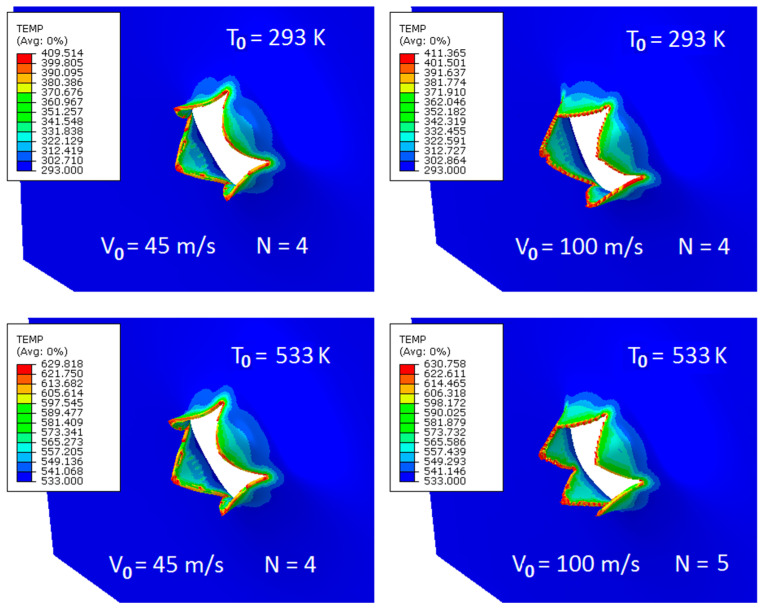
Abaqus simulations—temperature records for two different impact velocities V_0_ and two initial temperatures T_0_; max ΔT^num^ = 118 K at T_0_ = 293 K, max ΔT^num^ = 98 K at T_0_ = 533 K.

**Figure 10 materials-13-05821-f010:**
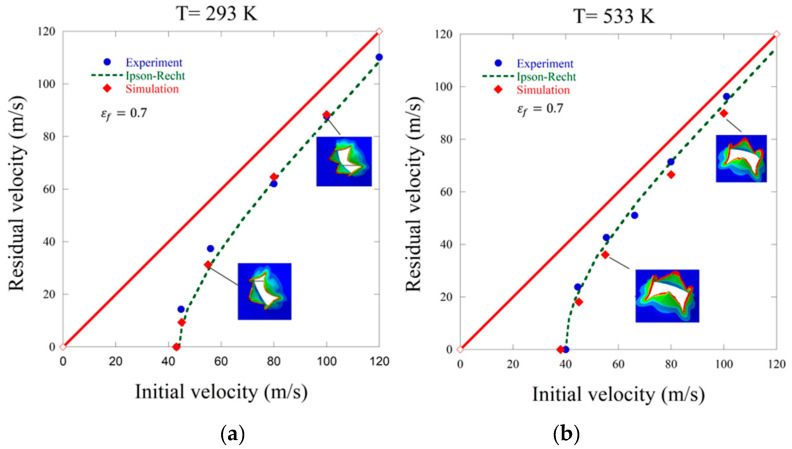
Initial impact velocity V_0_ vs. residual velocity V_R_: analytical, experimental and numerical results, (**a**) T_0_ = 293 K (20 °C), (**b**) T_0_ = 533 K (260 °C); numerical simulations with damage initiation equivalent plastic strain of $\varepsilon_{f}^{\mathrm{pl}}=\;0.7$; the Ipson–Recht model parameters were κ = 1.8 and V_B_ = 44 m/s for the temperature of T_0_ = 293 K and κ = 2.2 and V_B_ = 40 m/s for T_0_ = 533 K.

**Figure 11 materials-13-05821-f011:**
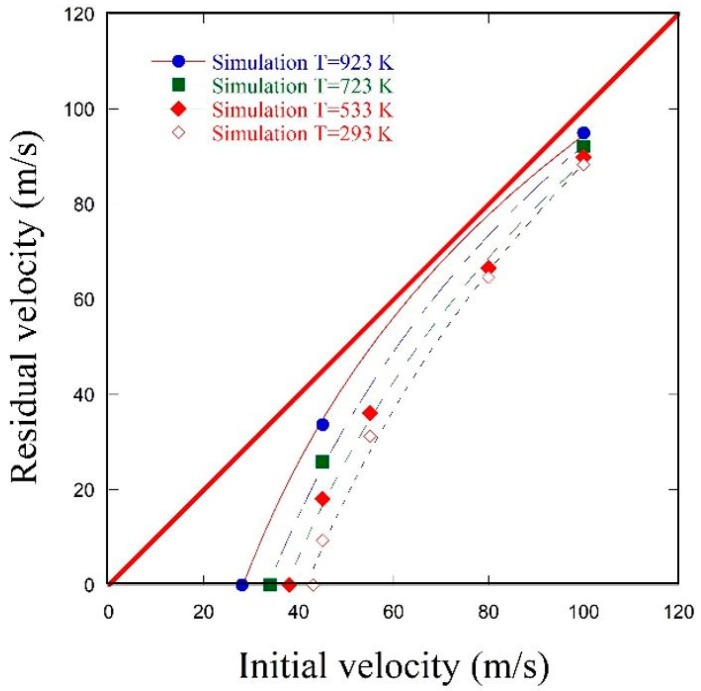
Evolution of initial impact velocity V_0_ vs. residual velocity V_R_ curves for the whole range of the analysed initial temperatures T_0_—numerical results.

**Figure 12 materials-13-05821-f012:**
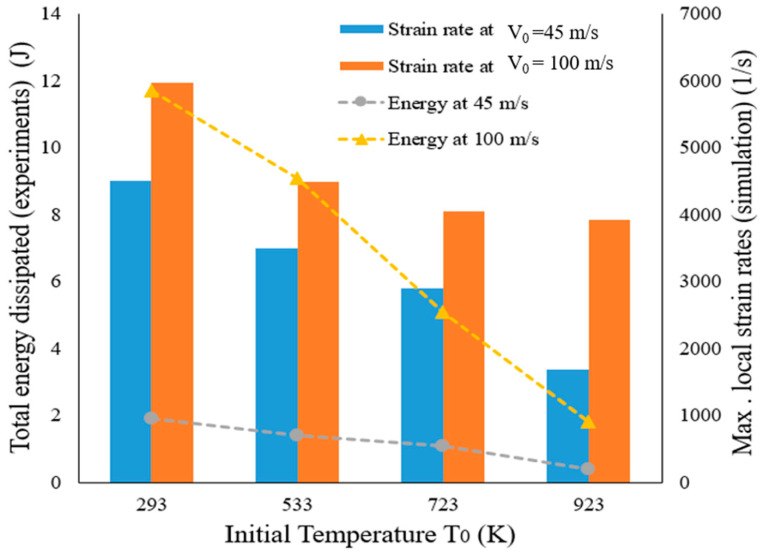
Evolution of the maximum local strain rates at most plastified elements and total energy dissipated during perforation process as a function of initial temperature T_0_ and initial impact velocity V_0_.

**Table 1 materials-13-05821-t001:** Summary of the perforation tests [9].

T = 293 K		T = 533 K		
V_0_ (m/s)	V_R_ (m/s)	V_0_ (m/s)	V_R_ (m/s)
43.00	0	40.00	0
44.74	14.37	44.54	23.83
55.93	37.45	55.37	42.68
65.28	48.84	66.14	51.11
80.02	62.11	79.87	71.43
100.00	87.75	101.01	96.34
119.91	110.22	121.07	113.64

NT.: values of V_0_ and V_R_ are mean values of 2–3 experiments per one initial impact velocity.

**Table 2 materials-13-05821-t002:** Material parameters for Johnson–Cook model.

A (MPa)	B (MPa)	n (-)	C (-)	m (-)	T_0_ (K)	T_m_ (K)	${\dot{\varepsilon }}_{0}\;(1/s)$
190	495.2	0.54	0.0021	1.45	293.15	1203.15	1.0

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
