# Peer review of "Mechanical Properties of Brass under Impact and Perforation Tests for a Wide Range of Temperatures: Experimental and Numerical Approach"

_materials, 2020, doi:10.3390/ma13245821_

Round 1

Reviewer 1 Report

The methodology follows is so interesting. However, I recommend authors to check the writing. The absence of conjunctive adverbs makes the document difficult to read in some parts. Some of the changes that are required are listed below:

  1. Abstract: it is difficult to understand which things belong to the previous study presented in reference [1], and which are original work in this paper. I would recommend starting with a sentence like: “Original perforation tests were performed in [1]”, highlighting all the aspect of these type of studies that the authors consider important. Then, the authors can continue: “in this work, these experiments were extended by compression and tension dynamic tests, in order to fully characterize the material tested, and a numerical model is presented to developed numerical simulation to reproduce perforation test”.
  2. Abstract: It is not clear which range of temperatures is used for the experimental tests and which one for the numerical simulations.
  3. Abstract: Line 28: “The ductile damage initiation criterion was used with plastic equivalent strain; a parametric study was carried out in order to define its form.” This sentence needs to be rewritten.
  4. Introduction: Line 44: “Dynamic tests are rarely coupled with thermal analysis since it is not easy to reach uniform temperature in the plate and gas gun are not frequently coupled to a thermal chamber.” What are the difficulties associated with the use of a thermal chamber and a gas gun together? It would be worthy to highlight this point.
  5. Introduction: Line 49: “Many authors dealt with perforation analysis, from theoretical approaches such as discussed in [2-3] to more practical considerations as reported in [4-7]”. What are these practical considerations?
  6. Introduction: Why is the line 62 in another paragraph?
  7. Introduction: what is an efficient constitutive relation?
  8. Introduction: I recommend change “deducted” by “calibrated”.
  9. Introduction: Could the authors describe the parametrical study?
  10. Experimental approaches: Line 70. Reading this sentence it seems that the main objective of this work was to develop the dynamic perforation tests, and these test were carried out in other work.
  11. Tension under quasi-static conditions: what machine was used for these tests? Describe the experimental procedure.
  12. Tension under quasi-static conditions: Line 79. change “is diminished” for “decreases”.
  13. Compression under dynamic loading using sandwich configuration: Line 87. change “three” for “four”.
  14. Compression under dynamic loading using sandwich configuration: rewrite sentence line 90: “On the other hand, it is visible that the failure at lower values of plastic strains for lower impact velocities”.
  15. Compression under dynamic loading using sandwich configuration: Figure1, it is difficult to see all the curves.
  16. Compression under dynamic loading using sandwich configuration: paragraph described in lines 104-106 must be before line 87.
  17. Perforation tests: Figure 5 appears in the test before Figure 4. Exchange them.
  18. Perforation test: line 133. “heated up in the same time”, change in for at.
  19. Perforation test: line 139. “Then, a special calibration specimen with the thermocouple is used to measure the temperature evolution in the specimen itself”. Could the authors explain the next sentence better?
  20. Perforation test: line 168. Could the authors confirm if T is the temperature?
  21. Modelling of the thermoviscoplastic behaviour: Could the authors describe the procedure follow to calibrate the material parameters?
  22. Modelling of the thermoviscoplastic behaviour: In line 198: “As demonstrated by the closer analysis, the strain rate and temperature effect on the failure criterion is marginal within the studied range of strain rates and temperatures, therefore a simplified  form was adopted” the authors say they have adopted a simplified form for the failure criterion, which one?  
  23. Numerical simulations of perforated test: Line 224: “modelled using the Johnson Cook model coupled to the Johnson Cook plasticity” I think there is a typo in this sentence.

Reviewer 2 Report

The authors presented the article "Mechanical properties of brass under impact and perforation tests for a wide range of temperatures: experimental and numerical approach". The reviewed article is of great cognitive and application significance. Figures, tables and terminology are clear and precise. Experimental research and numerical analysis are well described. The results were used to develop an effective model of yield stress for brass specimen.
The article is of a good scientific standard. 
I did not find any errors and therefore, in my opinion, it should be published in its current form.

Author Response

Dear Professor,

thank you for your kind review report. I am glad our work has been assessed in the way to be suitable for publishing in Materials.

Folowing the remarks given by other Reviewers, I have implemented some changes and made some improvements. I believe they will only improve the overall quality of the paper.

Kind regards,

Maciej Klósak

Reviewer 3 Report

The impact experiments perforation tests of brass were conducted and Johson-Cook constitutive equation was established. Based on the materials constitutive data, the FE simulations for impact and perforation were run and verified by the experiments. This research is deep and systemic, and the results are inspiring. The authors should improve their manuscript as follow:
1) The abstract is tedious and should be condensed.
2) The measured temperature results (temperature-time curves) are lacking. The adiabatic temperature arising from the large deformation may increase the pre-set temperature. It leads to non-constant temperature conditions during the impact.
3) On Abaqus, the impact was simulated. The damage evolution model and parameters (sub-option) should be provided.

Round 2

Reviewer 1 Report

The authors have made all suggested changes. So I consider it can be publish in the present form. However, I suggest them some some proofs:

  1. Introduction: The introduction starts with “The original perforation tests were performed in [1].” The authors should describe a little be more this test? and “Then a numerical model was presented to carry out numerical simulations”. The authors should specific that the objective of these numerical simulation? For example, to extend the study.
  2. Introduction: “Many authors dealt with perforation analysis, from theoretical approaches such as discussed in [2-3] to more practical considerations as reported in [4-9] where numerical applications are discussed for a variety of materials like metals or concrete.” The authors should specify more this sentence.
  3. Line 334: ”The next paragraph describes the equation used in numerical simulations”. Use section instead of paragraph.  
  4. Line 351: “The constants adopted are presented in Table 2.” Use parameters instead constants.
  5. Line 431: “The specimen is modelled using the model coupled to the Johnson Cook plasticity and assuming an isotropic behaviour.” What model do the authors refer to? The numerical model?
  6. Line 459: “The next chapter will compare numerical findings with experimental results of the perforation tests [1].” Use section instead of chapter.